# A Randomised Trial to Optimise Gestational Weight Gain and Improve Maternal and Infant Health Outcomes through Antenatal Dietary, Lifestyle and Exercise Advice: The OPTIMISE Randomised Trial

**DOI:** 10.3390/nu11122911

**Published:** 2019-12-02

**Authors:** Jodie M. Dodd, Andrea R. Deussen, Jennie Louise

**Affiliations:** 1Discipline of Obstetrics & Gynaecology, and Robinson Research Institute, The University of Adelaide, Adelaide, SA 5006, Australia; andrea.deussen@adelaide.edu.au (A.R.D.); jennie.louise@adelaide.edu.au (J.L.); 2Department of Perinatal Medicine Women’s and Children’s Hospital, North Adelaide, Adelaide, SA 5006, Australia

**Keywords:** dietary and lifestyle intervention, gestational weight gain, randomised controlled trial

## Abstract

There are well-recognised associations between excessive gestational weight gain (GWG) and adverse pregnancy outcomes, including an increased risk of pre-eclampsia, gestational diabetes and caesarean birth. The aim of the OPTIMISE randomised trial was to evaluate the effect of dietary and exercise advice among pregnant women of normal body mass index (BMI), on pregnancy and birth outcomes. The trial was conducted in Adelaide, South Australia. Pregnant women with a body mass index in the healthy weight range (18.5–24.9 kg/m^2^) were enrolled in a randomised controlled trial of a dietary and lifestyle intervention versus standard antenatal care. The dietitian-led dietary and lifestyle intervention over the course of pregnancy was based on the Australian Guide to Healthy Eating. Baseline characteristics of women in the two treatment groups were similar. There was no statistically significant difference in the proportion of infants with birth weight above 4.0 kg between the Lifestyle Advice and Standard Care groups (24/316 (7.59%) Lifestyle Advice versus 26/313 (8.31%) Standard Care; adjusted risk ratio (aRR) 0.91; 95% confidence interval (CI) 0.54 to 1.55; *p* = 0.732). Despite improvements in maternal diet quality, no significant differences between the treatment groups were observed for total GWG, or other pregnancy and birth outcomes.

## 1. Introduction

Obesity represents a significant global health burden, with the World Health Organisation highlighting the importance of weight gain prevention in adults of healthy weight, particularly among women of reproductive age [1]. In any given 5-year period, 20% of women of reproductive age have sufficient weight gain to progress them into a higher body mass index (BMI) category [2,3]. Furthermore, the rate of weight gain is highest (approximately 700 g per year) among women of normal BMI [4,5]. Pregnancy often represents a significant turning point in a woman’s cardiovascular and metabolic health trajectory secondary to pregnancy-related changes, including relative insulin resistance, which promotes weight gain [6], and risk of developing obesity subsequently [7,8].

There is substantial observational literature relating to gestational weight gain (GWG), which has been summarised by the Institute of Medicine (IoM) [9,10]. These recommendations advocate a GWG of 11.5 to 16.0 kg for women of normal body mass index (BMI 18.5–24.9 kg/m^2^) [9,10]. However, approximately 40% of women gain in excess of this amount [10]. There are well-recognised associations between excessive GWG and adverse pregnancy outcomes for the woman, including an increased risk of pre-eclampsia, gestational diabetes and caesarean birth [11,12,13,14]. There are also longer-term health consequences for women, including post-partum weight retention (PPWR) and development of obesity [15,16,17], with over 70% of women of normal BMI retaining more than 5 kg of weight 1 year after birth [18,19]. Women with excessive GWG also have a greatly increased risk of developing both diabetes [20,21] and cardiovascular disease in later life [22,23,24,25].

Excessive GWG is a well-recognised risk factor for high infant birth weight and is independently associated with an increased risk of child obesity in the offspring [26,27,28], thereby creating a vicious cycle in which the intergenerational effects of obesity are perpetuated [29]. Specifically, the risk of early childhood obesity increases by a factor of 1.08 (95% CI 1.03–1.14) per kilogram of maternal weight gained above the IoM recommendations [30]. Furthermore, it has been suggested that high maternal weight gain may induce a persisting susceptibility of an individual to an obesogenic environment [30,31]. There is also increasing evidence for an effect of excessive maternal GWG on subsequent cardiovascular risk and hypertension in children [27,32] and adolescents [33].

Despite recognition of the associations between excessive GWG in women of normal BMI during pregnancy and beyond, there is more limited information describing effective antenatal dietary interventions to optimise weight gain and improve health. In a systematic review of the literature, 12 randomised trials involving 2713 pregnant women were identified [34]. Of these trials, 8 specifically recruited 1048 women of normal BMI, although only 5 reported clinical outcomes (714 women) [34]. Providing a combined dietary and lifestyle intervention during pregnancy was associated with a modest 1.25 kg difference in weight gain (5 studies, 714 women) [34]. However, the effect on clinical pregnancy outcomes was less clear, being reported in only 2 trials, with 243 women [34].

The aim of the OPTIMISE randomised trial was therefore to evaluate the effect of dietary and exercise advice among pregnant women of normal BMI, on pregnancy and birth outcomes.

## 2. Materials and Methods

### 2.1. Trial Design

We conducted a randomised controlled trial, in which women with a BMI of 18.5 to 24.9 kg/m^2^, and a singleton pregnancy between 10 + 0–20 + 0 weeks were eligible to participate [35]. Women with a multiple pregnancy, or with diabetes (type 1 or type 2) diagnosed prior to pregnancy were excluded. Ethical approval was provided by the research ethics committee of the Women’s and Children’s Hospital (Adelaide, South Australia), approval number HREC/13/WCHN/152, and the study registered with the Australian and New Zealand Clinical Trials Registry (ACTRN 12614000583640). Recruitment to the trial commenced in June 2014.

Women were screened for eligibility at the time of their first antenatal appointment. All women presenting to the Women’s and Children’s Hospital had their height and weight measured, and BMI calculated by research staff. Eligible women were provided with information about the study and were counselled by a research assistant, prior to their provision of written consent to participate.

Randomisation: We used a computer-based randomisation service in the Discipline of Obstetrics and Gynaecology, The University of Adelaide. The randomisation schedule used balanced variable blocks with stratification for parity (0 versus 1 or more) and was prepared by an investigator who was not involved with recruitment or clinical care.

Women were randomised to either the ‘Lifestyle Advice Group’ or the ‘Standard Care Group’. Blinding of participants was not possible given the nature of the intervention, but where possible, antenatal care-providers, outcome assessors and data analysts were blinded to treatment allocation.

Treatment Allocation: Women randomised to the Lifestyle Advice Group received an intervention consisting of six sessions provided across the course of pregnancy. Three sessions were face-to-face, with two provided by the dietitian shortly after trial entry and again at 28 weeks’ gestation, and one provided by a research assistant at 36 weeks’ gestation. Women also received three telephone calls from the research assistant at 20, 24 and 32 weeks’ gestation. The dietary advice provided was consistent with current Australian dietary standards [36], while specifically maintaining a balance of carbohydrates, fat and protein, and encouraging women to reduce their intake of energy dense and non-core foods high in refined carbohydrates and saturated fats. Women were advised to increase their intake of fibre, and to consume two servings of fruit, five servings of vegetables and three servings of dairy each day [36,37,38].

Tailoring of the intervention was informed by stage theories of health decision making where an individual progresses through a series of cognitive phases when undertaking behavioural change [39]. The initial planning session with a research dietitian provided women with written dietary and activity information, an individual diet and physical activity plan, recipe book and example menu plans. Women were encouraged to set achievable goals for dietary and exercise change, supported to make these lifestyle changes and to self-monitor their progress, using a SMART goals approach. The SMART Goal approach includes setting goals that are specific, measurable, achievable, realistic and timely. Therefore, a SMART goal incorporates all of these criteria to increase the chances of goal achievement. These principles were reinforced at subsequent contacts with research staff [37,38].

Women who were randomised to the Standard Care Group received their antenatal care according to hospital guidelines, which did not include information relating to dietary intake, physical activity or weight gain during pregnancy.

All women were asked to complete a food frequency questionnaire, physical activity questionnaire and quality of life assessments at trial entry, 28 and 36 weeks’ gestation and six months postpartum. Each woman’s weight was recorded at trial entry and at 36 weeks’ gestation or nearest to birth, with gestational weight gain determined as the difference between weight at 36 weeks and trial entry. All women were offered a research ultrasound at 28 and 36 weeks’ gestation to assess foetal growth (results not presented in this manuscript). After birth, information relating to birth and infant outcomes was obtained from the case notes by the research assistant, who remained blinded to the woman’s allocated treatment group.

Consistent with state-wide clinical practice guidelines, all women were screened for gestational diabetes at approximately 28 weeks’ gestation [40]. During the course of the trial, diagnostic criteria for gestational diabetes changed across the state from a positive 75 g oral glucose tolerance test with fasting blood glucose > 5.5 mmol/L, or 2 h ≥ 7.8 mmol/L, to fasting blood glucose ≥ 5.1 mmol/L, 1 h ≥ 10.0 mmol/L, or 2 h ≥ 8.5 mmol/L [40]. Women diagnosed with gestational diabetes remained in the study and were offered treatment with further dietary modification and metformin or insulin added as required to maintain appropriate glycaemic control [40].

### 2.2. Outcome Measures

The primary trial outcome was the proportion of infants with birth weight > 4 kg. A range of secondary study outcomes were collected and listed below.

#### 2.2.1. Secondary Infant Outcomes

Adverse outcomes for the infant: including preterm birth before 37 weeks’ gestation; perinatal mortality (either stillbirth (intrauterine foetal death after trial entry and prior to birth), or infant death (death of a live born infant prior to hospital discharge, and excluding lethal congenital anomalies)); infant birth weight; infant birth weight < 2500 g; infant birth weight > 4500 g; large for gestational age defined as infant birth weight > 90th percentile for gestational age and infant sex; small for gestational age defined as infant birth weight < 10th percentile for gestational age and infant sex; hypoglycaemia requiring intravenous treatment; admission to neonatal intensive care unit or special care baby unit; hyperbilirubinaemia requiring phototherapy; nerve palsy; fracture; birth trauma; shoulder dystocia; corticosteroid use; respiratory distress syndrome (with moderate or severe respiratory disease defined as mean airway pressure > 10 cm H_2_O and/or inspired oxygen fraction (FiO_2_) > 0.80 with ventilation) [41]; discharge home on oxygen; patent ductus arteriosus; proven systemic infection requiring treatment; retinopathy of prematurity; necrotising enterocolitis; neonatal encephalopathy [42]; seizures; length of hospital stay; and infant not exclusively breast fed at hospital discharge.

#### 2.2.2. Maternal Antepartum, Labour and Birth Outcomes

Adverse outcomes for the woman: including maternal hypertension and pre-eclampsia (in accordance with recognised Australasian Society for the Study of Hypertension in Pregnancy criteria) [43]; maternal gestational diabetes; need for and length of antenatal hospital stay; antepartum haemorrhage requiring hospitalisation; preterm prelabour ruptured membranes; chorioamnionitis; need and reason for induction of labour; any antibiotic use during labour; caesarean section; postpartum haemorrhage (blood loss > 600 mL); perineal trauma; wound infection; endometritis; length of postnatal hospital stay; thromboembolic disease; maternal death.

#### 2.2.3. Maternal Weight Change

Maternal weight changes: including total gestational weight gain; average weekly gestational gain; gestational weight gain below, within and above IoM recommendations [10]; and anthropometric assessment (skin-fold thickness measurement (SFTM), body circumferences and bio-impedance to assess adiposity).

#### 2.2.4. Maternal Diet and Physical Activity 

Maternal changes in diet and physical activity as measured by questionnaires completed by the woman at trial entry, 28 and 36 weeks’ gestation (Harvard Semi-quantitative Food Frequency Questionnaire [44,45], and the Short Questionnaire to Assess Health-enhancing physical activity (SQUASH) [46]).

#### 2.2.5. Maternal Quality of Life

Maternal quality of life and emotional wellbeing as measured by questionnaires completed by the woman at trial entry, 28 weeks and 36 weeks’ gestation relating to quality of life (as measured using the SF12 Health Survey Questionnaire) [47]; preferences for treatment, satisfaction with care; anxiety (as measured by the Short Form Spielberger State Trait Inventory [48]) and depression (as measured by the Edinburgh Postnatal Depression Scale [49]).

### 2.3. Sample Size Estimate

The primary clinical endpoint was the incidence of infants born with birth weight > 4 kg, with an estimated incidence in women eligible for this trial of 8.72% [50]. To detect a difference from 8.72% to 3.89% (alpha 0.05; power 70%), we required 624 women.

### 2.4. Analysis and Reporting of Results

All analyses followed a pre-specified statistical analysis plan, as shown in Appendix A. Baseline characteristics of all randomised women were examined descriptively as an indication of comparable treatment groups, and included maternal age, parity, race, height, weight, smoking status, past obstetric history and a diagnosis of previous gestational diabetes. Primary and secondary outcomes were analysed on an “intention to treat” basis, according to the treatment allocated (Lifestyle Advice or Standard Care) at the time of randomisation. Continuous outcomes were analysed using linear regression, and binary outcomes were analysed using log binomial regression. Outcomes measured at multiple time points included a time-by-treatment interaction term, with generalised estimating equations used to account for correlation between repeated measures.

As specified in the Statistical Analysis Plan, the primary analyses were adjusted analyses based on imputed data. Unadjusted analyses, and analyses on unimputed data (not presented), were also performed as secondary sensitivity analyses. Adjusted models included the stratification variable (parity) as well as BMI (continuous variable), smoking, socio-economic status (as indicated by the Australian Bureau of Statistics’ 2011 Socio-economic Index for Areas—Index of Relative Socio-economic Disadvantage (SEIFA IRSD) quintile) and maternal age at trial entry as covariates.

There was one missing value for the primary outcome and other infant birth weight outcomes; many other outcomes (including infant anthropometry, infant and maternal delivery data) had less than 1% missing data, while infant SFTM and other maternal antenatal measures had between 20–40% missing data. Multiple imputation by the fully conditional specification (chained equations) method was used to create 100 complete datasets for analysis [51]. The imputation model included all outcomes, all stratification variables, maternal baseline height, weight and gestational age, and maternal weight at 36 weeks’ gestation. Estimates were derived in the standard manner by combining the estimates from each imputation using Rubin’s rules [51]. As there was only one missing value for the primary outcome, no missing not at random (MNAR) sensitivity analyses were performed. Analyses were performed using Stata version 15 (StataCorp, 77845 Texas, USA).

Data cannot be made publicly available because of ethics and Institutional Review Board restrictions. However, researchers can apply for data access to the corresponding author.

## 3. Results

### 3.1. Participant Characteristics

Between June 2014 and April 2017, 2602 eligible women were approached to participate, with 645 randomised, 323 (50.1%) to Lifestyle Advice and 322 (49.9%) to Standard Care, as shown in Figure 1. Four women were randomised in error prior to the start of the trial (all four from the Lifestyle Advice Group) and were not included in analyses, leaving a total of 641 women (319 Lifestyle Advice Group and 322 Standard Care). A further four women (two in each group) terminated pregnancies for foetal anomalies; four women suffered a stillbirth (all in the Standard Care Group); and two liveborn infants died after birth (both in the Lifestyle Advice Group). Two stillbirths occurred in the setting of chorioamnionitis prior to 24 weeks’ gestation; one occurred at 40 weeks secondary to *Escherichia coli* sepsis; and one unexplained stillbirth occurred at 39 weeks. One liveborn infant died secondary to extreme prematurity following spontaneous birth at 23 weeks, and the second infant, born at 36 weeks, died at a few hours of age from pulmonary hypoplasia secondary to multicystic dysplastic kidney disease. Overall, 633 women and 629 liveborn infants were included in the analyses, with adequate data available for 628 (99.8%) for the primary outcome of birth weight above 4.0 kg. There were no maternal deaths.

The baseline characteristics of women in the two treatment groups were similar at trial entry, as shown in Table 1. The median BMI of the cohort was 22.20 kg/m^2^ (inter-quartile range (IQR) 20.87 to 23.60 kg/m^2^). The mean maternal age of participants was 31.5 years, with 59% of women in their first ongoing pregnancy. The median gestational age at trial entry was approximately 16.3 weeks (IQR 14.57 to 18.14 weeks), 4.4% of women were smokers, and 30.5% of women were from the highest two quintiles of social disadvantage.

### 3.2. Pre-Specified Infant Outcomes

There was no statistically significant difference in the proportion of infants with birth weight above 4.0 kg between the Lifestyle Advice and Standard Care groups (24/316 (7.59%) Lifestyle Advice versus 26/313 (8.31%) Standard Care; adjusted risk ratio (aRR) 0.91; 95% confidence interval (CI) 0.54 to 1.55; *p* = 0.732), as shown in Table 2. 

### 3.3. Maternal Diet Quality

When compared with women who received standard care, women who received lifestyle advice demonstrated improvements in their reported dietary quality as measured by the healthy eating index (HEI) at both 28 (74.35 ± 7.65 Lifestyle Advice Group vs. 72.11 ± 8.21 Standard Care Group; adjusted mean difference 2.21; 95% CI 0.98 to 3.45; *p* < 0.001) and 36 weeks’ gestation (74.10 ± 8.77 Lifestyle Advice Group vs. 72.50 ± 8.43 Standard Care Group; adjusted mean difference 1.57; 95% CI 0.22 to 2.91; *p* = 0.023), as shown in Table 3. There were no observed differences in reported physical activity, as shown in Table 3.

### 3.4. Pre-Specified Maternal Antepartum Outcomes

Despite improvements in maternal diet quality, there were no differences between the treatment groups observed for total GWG (11.32 ± 3.96 kg Lifestyle Advice versus 11.70 ± 3.78 kg Standard Care; adjusted mean difference (aMD)—0.37; 95% CI—0.97 to 0.23; *p* = 0·227), as shown in Table 4. Similarly, there were no observed differences in the proportion of women who gained weight above (28 (8.72%) Lifestyle Advice versus 41 (13.16%) Standard Care; aRR 0.58; 95% CI 0.32 to 1.04; *p* = 0.066) or below (160 (50.71%) Lifestyle Advice versus 162 (51.68%) Standard Care; aRR 0.85; 95% CI 0.60 to 1.21; *p* = 0.366) the IOM recommendations, as shown in Table 4.

There were no significant differences observed between the two treatment groups with regards to the occurrence of pregnancy-related complications, including hypertension, pre-eclampsia or eclampsia, gestational diabetes, antepartum haemorrhage or preterm prelabour ruptured membranes, as shown in Table 4. There were no significant differences in the number of antenatal hospital admissions between the two groups, however, there was a borderline statistically significant difference in the mean number of antenatal days in hospital (0.83 (4.18) Lifestyle Advice Group versus 0.42 (1.49) in the Standard Care Group; aRR 1.99; 95% CI 1.03 to 3.85; *p* = 0.042). This difference can be explained by the extremely long antenatal hospital admissions for three women in the Lifestyle Advice Group, as shown in Table 4. Self-reported maternal quality of life was similar between groups, as shown in Appendix A.

### 3.5. Pre-Specified Maternal Labour and Birth Outcomes

While there was a reduction in the risk of requiring induction of labour among women in the Lifestyle Advice group (74 (23.42%) Lifestyle Advice versus 109 (34.96%) Standard Care; aRR 0.66; 95% CI 0.52 to 0.85; *p* = 0·001), this likely reflected chance differences in the need for induction of labour for post-dates pregnancy (21 (28.38%) Lifestyle Advice versus 43 (39.45%) Standard Care), as shown in Table 5. There were no significant differences observed between the two groups with regards to risk of caesarean birth (73 (23.17%) Lifestyle Advice versus 74 (23.79%) Standard Care; aRR 0.95; 95% CI 0.72 to 1.26; *p* = 0.713).

The mean gestational age at birth was lower in the Lifestyle Advice Group (39.12 ± 2.38 weeks Lifestyle Advice versus 39.46 ± 1.63 weeks Standard Care; aMD—0.34; 95% CI—0.66 to −0.02; *p* = 0·039), as shown in Table 2. While this difference is statistically significant, the difference is considered clinically small, and reflective of the differences in induction of labour for post-dates pregnancy. The non-statistically significant difference observed in mean infant birth weight (3291.97 ± 586.20 g Lifestyle Advice versus 3370.92 ± 511.24 g Standard Care; aMD—78.39; 95% CI—164.00 to 7.22; *p* = 0.073) is also reflective of the observed difference in mean gestational age at birth; the mean difference in birthweight z-score was not near statistical significance (aMD—0.04; 95% CI—0.18, 0.09, *p* = 0.532). There were no statistically significant differences between the two groups with regards to other infant outcomes, as shown in Table 2, or newborn anthropometric measures, as shown in Appendix A.

### 3.6. Effect Modification by Maternal Pre-Pregnancy BMI

Pre-specified secondary analyses identified some evidence of effect modification by maternal pre-pregnancy BMI, suggesting that the intervention may have been more effective in women with higher maternal BMI in reducing infant birth weight and head, abdominal and chest circumferences, as shown in Appendix A. There was also some weak evidence suggesting a differential effect of the intervention by parity on infant birth weight, chest and arm circumference and thigh skinfold measurement, with lifestyle advice being more effective in reducing these measures among women in their second and subsequent pregnancy.

## 4. Discussion

Our findings indicate that providing lifestyle advice during pregnancy to women with BMI within the normal range was associated with improvements in maternal diet quality over the course of pregnancy. However, despite improvements in maternal diet, lifestyle advice was not associated with any differences in total gestational weight gain or risk of weight gain below or above the IOM recommendations. There were no significant differences in clinical outcomes for either women or their infants, including risk of infant birth weight above 4 kg.

There were a number of strengths to our randomised trial. To our knowledge, it was the largest of its kind recruiting women with healthy BMI during pregnancy, with comprehensive reporting of relevant maternal and infant clinical outcomes, high rates of participant follow-up and broad inclusion criteria. Our methodology was robust, with all participating women prospectively having their height, weight and BMI measured, use of a central randomisation service and outcome assessors who were blinded to the woman’s allocated treatment group. Furthermore, both the content and intensity of the intervention reflect one that could be realistically achieved within current public antenatal care services.

Participants in our trial were predominantly white Caucasian, with less than half of women from areas of high social disadvantage. Furthermore, 75% of eligible women declined participation due to time constraints, lack of interest or lack of perceived need. These factors may limit our external validity and generalisability of our findings to other patient populations.

Our systematic review evaluating dietary and lifestyle interventions in pregnant women with healthy BMI, providing a combined intervention was associated with a modest 1.25 kg difference in weight gain (5 studies, 714 women) [34]. Overall, the methodological quality of the studies included were of medium to high quality, and low to medium risk of bias [34]. The intensity and nature of the intervention overall was poorly described, with nine interventions consisting of face-to-face sessions with a trained professional [52,53,54,55,56,57,58,59,60]. The intensity ranged from three dietetic sessions over pregnancy [58,59], up to one at each antenatal visit [61]. Three studies [52,54,58] provided an additional session post-partum.

The findings of our current study are in contrast to this review [34], finding no clinically or statistically significant difference in total gestational weight gain, or risk of weight gain below or above the IOM recommendations. However, provision of dietary and lifestyle advice during pregnancy was associated with improvements in maternal self-reported diet quality as measured by the HEI. These findings are consistent with those we have reported previously from both the LIMIT [37,38] and GROW [61] randomised trials, highlighting the reproducibility of the intervention among pregnant women across the BMI spectrum in effecting dietary change.

Overall, however, our findings are consistent with the broader literature describing antenatal dietary and lifestyle interventions in pregnant women across all BMI categories [62]. In a comprehensive individual participant data meta-analysis utilising data from 36 randomised trials, and more than 12,500 pregnant women, a modest effect on GWG was identified following dietary and physical activity advice (mean difference −0.7 kg), although there was very little effect on clinical pregnancy and neonatal outcomes [62]. When considered in their totality, the available literature challenges the current underlying rationale of providing an antenatal dietary and lifestyle intervention with the intention of limiting weight gain as a means to improving pregnancy outcomes. GWG reflects a combination of maternal fat deposition, pregnancy related plasma volume expansion, breast and uterine tissue hypertrophy, extracellular fluid, placental mass, foetal mass and amniotic fluid volume [63], and while it has been considered a surrogate for adiposity gain in pregnancy, the evidence to date suggests that it may not be readily modified simply through changes in maternal dietary intake and physical activity.

## 5. Conclusions

Our findings indicate that while providing lifestyle advice during pregnancy to women with BMI within the normal range was associated with improvements in maternal diet quality, there were no clinically or statistically significant differences in total gestational weight gain or in clinical outcomes for either women or their infants. Providing such an intervention in pregnancy is not advocated.

## Figures and Tables

**Figure 1 nutrients-11-02911-f001:**
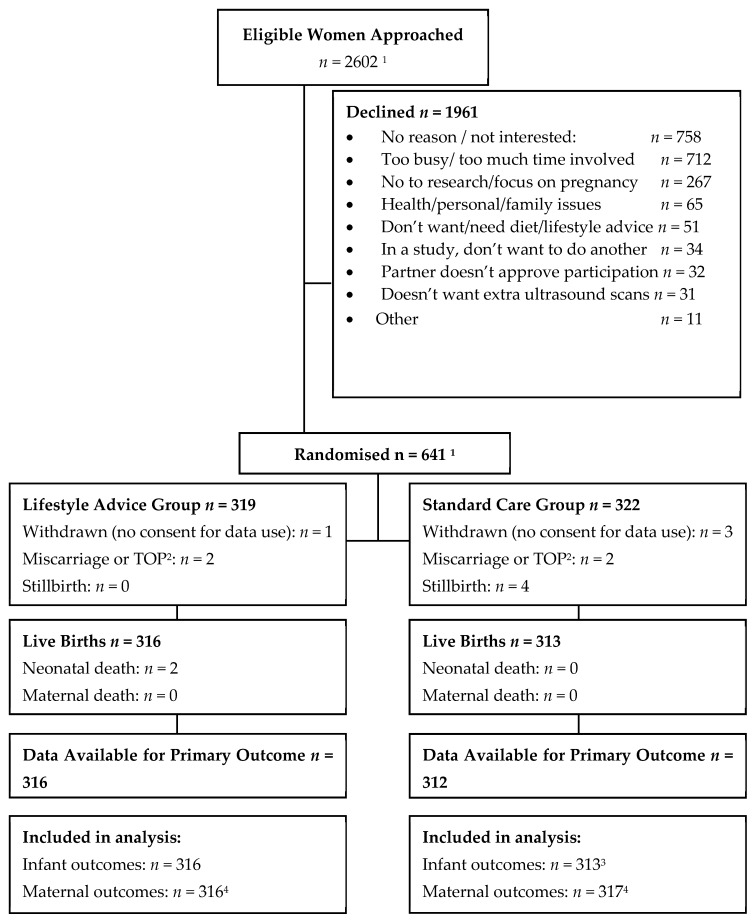
Flow of participants in the trial. Notes: ^1^ This number excludes four women who were randomised in error prior to trial registration. ^2^ Termination of pregnancy (TOP) ^3^ Three hundred and twelve infants with non-missing data included in raw data analysis, one infant with missing data had outcomes imputed and was therefore included in the imputed analysis. ^4^ Stillbirths excluded from infant outcomes analysis but included for analysis of maternal antenatal outcomes only.

**Table 1 nutrients-11-02911-t001:** Baseline characteristics.

Characteristic	Lifestyle Advice (*n* = 316) **	Standard Care (*n* = 317) **	Overall (*n* = 633) **
Maternal age in (years) *	31.60 (4.63)	31.45 (4.63)	31.53 (4.76)
Gestational age at entry (weeks) ^+^	16.21 (14.43, 18.14)	16.29 (14.71, 18.14)	16.29 (14.57, 18.14)
Body mass index at entry (kg/m^2^) ^+^	22.17 (20.81, 23.70)	22.20 (20.90, 23.46)	22.20 (20.87, 23.60)
Height at trial entry ^+^	165.18 (7.18)	164.74 (7.18)	164.96 (7.17)
Weight at trial entry ^+^	60.56 (6.92)	60.22 (6.92)	60.39 (6.88)
Public patient ^#^	312 (98.73)	315 (99.37)	627 (99.05)
Ethnicity ^#^			
Caucasian	212 (67.09)	215 (67.82)	427 (67.46)
Asian	50 (15.82)	45 (14.20)	95 (15.01)
Indian, Pakistani, Sri Lankan	22 (6.96)	29 (9.15)	51 (8.06)
Other	32 (10.13)	28 (8.83)	60 (9.47)
Nulliparous ^#^	189 (59.81)	186 (58.68)	375 (59.24)
Smoker ^#^	15 (4.75)	13 (4.10)	28 (4.42)
SEIFA IRSD ^1^ Quintile ^# ^^			
Q1 (most disadvantaged)	48 (15.19)	58 (18.30)	106 (16.75)
Q2	78 (24.68)	93 (29.34)	87 (13.74)
Q3	48 (15.19)	39 (12.30)	156 (24.64)
Q4	80 (25.32)	76 (23.97)	156 (24.64)
Q5 (least disadvantaged)	62 (19.62)	51 (16.09)	113 (17.85)

* = mean and standard deviation. ^+^ = median and interquartile range. ^#^ = number and %. ^ = socioeconomic index of relative social disadvantage as measured by SEIFA. ** = includes all women randomised who did not withdraw consent to use their data, and who did not suffer miscarriage or termination of pregnancy prior to 20 weeks gestation. Note that numbers reported in this table incorporate some corrections post-randomisation to parity categories used (as stratification variable) when randomising participants. Eighteen participants were categorised as having parity 0 at randomisation when they in fact had parity 1+, and five participants were categorised as having parity 1+ at randomisation when they in fact had parity 0. ^1^ SEIFA IRSD: Socio-economic Index for Areas—Index of Relative Socio-economic Disadvantage.

**Table 2 nutrients-11-02911-t002:** Pre-specified infant outcomes by treatment group.

Outcome	Lifestyle Advice (*n* = 316) **	Standard Care (*n* = 313) **	Unadjusted Estimate (95% CI)	Unadjusted *p* Value	Adjusted Estimate (95% CI) ^c^	Adjusted *p* Value
Birthweight > 4 kg ^a^	24 (7.59)	26 (8.31)	0.91 (0.54, 1.56)	0.739	0.91 (0.54, 1.55)	0.732
Birthweight (g) ^b^	3291.97 (586.20)	3370.92 (511.24)	−78.96 (−164.95, 7.03)	0.072	−78.39 (−164.00, 7.22)	0.073
Birthweight z-score ^b^	−0.01 (0.87)	0.04 (0.89)	−0.05 (−0.18, 0.09)	0.503	−0.04 (−0.18, 0.09)	0.532
Gestational age at delivery (weeks) ^b^	39.12 (2.38)	39.46 (1.63)	−0.33 (−0.65, −0.02)	0.040	−0.34 (−0.66, -0.02)	0.039
Large for gestational age ^a^	22 (6.96)	25 (8.00)	0.87 (0.50, 1.51)	0.621	0.88 (0.51, 1.52)	0.641
Small for gestational age ^a^	21 (6.65)	25 (8.01)	0.83 (0.47, 1.45)	0.512	0.84 (0.48, 1.47)	0.545
Birthweight below 2.5kg ^a^	20 (6.33)	15 (4.82)	1.31 (0.69, 2.52)	0.411	1.32 (0.69, 2.54)	0.399
Birthweight > 4.5 kg ^d^	0 (0.00)	2 (0.64)		0.246		
Neonatal intensive/special care nursery admission ^a^	27 (8.54)	34 (10.89)	0.78 (0.49, 1.27)	0.323	0.80 (0.50, 1.30)	0.368
Neonatal death ^d^	2 (0.63)	0 (0.00)		0.499		
Hypoglycaemia ^a^	10 (3.16)	23 (7.41)	0.43 (0.21, 0.88)	0.022	0.44 (0.21, 0.91)	0.026
Hyperbilirubinaemia ^a^	23 (7.28)	15 (4.80)	1.52 (0.81, 2.85)	0.196	1.53 (0.82, 2.87)	0.181
Shoulder dystocia ^a^	9 (2.85)	13 (4.16)	0.68 (0.30, 1.58)	0.373	0.69 (0.30, 1.59)	0.386
Nerve palsy ^a, d^	0 (0.00)	0(0.00)	--		--	
Bone fracture ^a, d^	0 (0.00)	1 (0.32)	--	0.497	--	
Birth trauma ^a, d^	1 (0.32)	2 (0.64)	--	0.622	--	

^a^ Number and percentage, estimates are relative risks and 95% confidence interval. ^b^ Mean and standard deviation, estimates are differences in means and 95% confidence interval. ^d^ Due to small numbers of events, no analysis of these outcomes was possible; the p value for comparison between groups has been calculated using a Fisher’s Exact Test. For nerve palsy, there were no events in either group. ^c^ Adjusted analyses: all outcomes were adjusted for the stratification variable parity (0 vs. 1+), maternal age (continuous), maternal pre-pregnancy body mass index (BMI) (continuous) and SEIFA IRSD Quintile. ** = includes all infants of women randomised who did not withdraw consent to use their data, and who did not suffer miscarriage or termination of pregnancy prior to 20 weeks gestation, or stillbirth.

**Table 3 nutrients-11-02911-t003:** Pre-specified maternal antepartum diet quality and physical activity outcomes by treatment group.

Outcome	Lifestyle Advice (*n* = 316) **	Standard Care (*n* = 313) **	Unadjusted Estimate (95% CI)	Unadjusted *p* Value	Adjusted Estimate (95% CI) ^f^	Adjusted *p* Value
Healthy Eating Index ^b, e^				<0.001 *		< 0.001 *
Trial Entry	72.94 (9.22)	73.56 (7.89)	−0.62 (−1.96, 0.72)	0.362	−0.66 (−1.99, 0.68)	0.334
28 Weeks	74.35 (7.65)	72.11 (8.21)	2.25 (1.01, 3.48)	<0.001	2.21 (0.98, 3.45)	< 0.001
36 Weeks	74.10 (8.77)	72.50 (8.43)	1.60 (0.25, 2.95)	0.020	1.57 (0.22, 2.91)	0.023
Total Energy (kJ) ^b, e^				0.017 *		0.017 *
Trial Entry	8917.75 (3182.65)	8899.67 (3796.04)	18.07 (−525.98, 562.13)	0.948	20.93 (−516.71, 558.57)	0.939
28 Weeks	9358.47 (3782.64)	8692.51 (2829.24)	665.95 (141.52, 1190.38)	0.013	668.81 (155.43, 1182.19)	0.011
36 Weeks	8809.72 (3233.50)	8697.78 (3132.76)	111.93 (−381.78, 605.65)	0.657	114.79 (−375.44, 605.03)	0.646
Glycaemic Index ^b, e^				0.183 *		0.183 *
Trial Entry	47.23 (4.85)	47.54 (4.88)	−0.31 (−1.07, 0.45)	0.427	−0.26 (−1.01, 0.49)	0.496
28 Weeks	47.22 (3.70)	48.13 (4.46)	−0.91 (−1.55, −0.27)	0.005	−0.86 (−1.50, −0.23)	0.008
36 Weeks	47.18 (4.73)	47.79 (4.57)	−0.61 (−1.34, 0.12)	0.099	−0.56 (−1.28, 0.16)	0.124
Glycaemic Load ^b, e^				0.113 *		0.113 *
Trial Entry	110.53 (48.61)	115.02 (67.51)	−4.50 (−13.76, 4.76)	0.341	−4.06 (−13.15, 5.02)	0.381
28 Weeks	117.43 (55.86)	114.87 (45.96)	2.56 (−5.43, 10.54)	0.530	2.99 (−4.89, 10.87)	0.457
36 Weeks	109.28 (46.41)	113.87 (51.31)	-4.59 (-12.14, 2.95)	0.233	−4.16 (−11.62, 3.31)	0.275
Metabolic Equivalent Task Score ^b, e^				0.998 *		0.998 *
Trial Entry	9809.81 (4176.78)	9744.88 (4427.83)	64.93 (−607.56, 737.42)	0.850	82.27 (−581.63, 746.16)	0.808
28 Weeks	9085.84 (4076.53)	9028.78 (4440.41)	57.06 (−614.93, 729.06)	0.868	74.40 (−588.00, 736.81)	0.826
36 Weeks	7863.59 (4848.39)	7786.89 (4609.72)	76.70 (−664.65, 818.05)	0.839	94.04 (−637.23, 825.32)	0.801

^b^ Mean and standard deviation, estimates are differences in means and 95% confidence interval. ^e^ Repeated measures outcomes: models included a time by intervention interaction term, and separate estimates of treatment effect were derived at each time point regardless of the significance of this interaction term. ^f^ Adjusted analyses: all outcomes were adjusted for the stratification variable parity (0 vs. 1+), maternal age (continuous), maternal pre-pregnancy BMI (continuous) and SEIFA IRSD Quintile. * Denotes p value testing for interaction between treatment and time, i.e., whether effect of intervention differed between time points. ** = includes all women randomised who did not withdraw consent to use their data, and who did not suffer miscarriage or termination of pregnancy prior to 20 weeks gestation, or stillbirth.

**Table 4 nutrients-11-02911-t004:** Pre-specified maternal antepartum outcomes by treatment group.

Outcome	Lifestyle Advice (*n* = 316) **	Standard Care (*n* = 313) **	Unadjusted Estimate (95% CI)	Unadjusted *p* Value	Adjusted Estimate (95% CI) ^f^	Adjusted *p* Value
Total Gestational Weight Gain (kg) ^b^	11.32 (3.96)	11.70 (3.78)	−0.39 (−0.99, 0.21)	0.205	−0.37 (−0.97, 0.23)	0.227
Average Weekly Gestational Gain (kg) ^b^	0.57 (0.21)	0.60 (0.21)	−0.03 (−0.06, 0.01)	0.114	−0.03 (−0.06, 0.01)	0.132
Institute of Medicine Category: total gestational weight gain ^d^				0.168		0.177^
Below	160 (50.71)	162 (51.68)	0.85 (0.60, 1.21)	0.362	0.85 (0.60, 1.21)	0.366
Within	128 (40.57)	110 (35.16)	(reference)		(reference)	
Above	28 (8.72)	41 (13.16)	0.57 (0.32, 1.03)	0.062	0.58 (0.32, 1.04)	0.066
Institute of Medicine Category: weekly gestational weight gain ^d^				0.386		0.444^
Below	61 (19.25)	46 (14.78)	1.40 (0.84, 2.31)	0.196	1.36 (0.82, 2.26)	0.235
Within	105 (33.12)	111 (35.47)	(reference)		(reference)	
Above	151 (47.63)	156 (49.75)	1.03 (0.71, 1.48)	0.895	1.02 (0.70, 1.48)	0.916
Pregnancy Hypertension ^a^	5 (1.58)	4 (1.30)	1.22 (0.33, 4.51)	0.764	1.87 (0.52, 6.70)	0.338
Pre-Eclampsia/Eclampsia ^a^	6 (1.90)	9 (2.91)	0.65 (0.24, 1.81)	0.414	0.70 (0.25, 1.96)	0.502
Clinical Diagnosis of Gestational Diabetes Mellitus ^a, †^	39 (12.43)	39 (12.46)	1.00 (0.64, 1.55)	0.995	1.02 (0.66, 1.59)	0.929
Antenatal Hospital Admission ^a^	43 (13.61)	52 (16.61)	0.82 (0.56, 1.19)	0.294	0.81 (0.56, 1.18)	0.272
Antenatal Length Stay ^c^	0.83 (4.18)	0.42 (1.49)	1.98 (1.00, 3.92)	0.049	1.99 (1.03, 3.85)	0.042
Antepartum Haemorrhage ^a^	4 (1.27)	7 (2.24)	0.57 (0.17, 1.91)	0.360	0.62 (0.18, 2.10)	0.443
Preterm Prelabour Ruptured Membranes ^a^	5 (1.58)	4 (1.28)	1.24 (0.34, 4.57)	0.748	1.17 (0.32, 4.32)	0.814

^a^ Number and percentage, estimates are relative risks and 95% confidence interval regression models. ^b^ Mean and standard deviation, estimates are differences in means and 95% confidence interval. ^c^ Mean and standard deviation, and estimates are relative risk ratios and 95% confidence intervals. ^d^ Number and percentage in each category, estimates are odds ratios and 95% confidence intervals. ^f^ Adjusted analyses: all outcomes were adjusted for the stratification variable parity (0 vs. 1+), maternal age (continuous), maternal pre-pregnancy BMI (continuous) and SEIFA IRSD Quintile. ^ Denotes p value for a global test of any difference between categories in the multinomial logistic regression model. ^†^ Adjusted model for this outcome required log Poisson regression with robust variance estimation due to convergence issues with the log binomial model. ** = includes all women randomised who did not withdraw consent to use their data, and who did not suffer miscarriage or termination of pregnancy prior to 20 weeks gestation, or stillbirth.

**Table 5 nutrients-11-02911-t005:** Pre-specified maternal labour and birth outcomes by treatment group.

Outcome	Lifestyle Advice (*n* = 316) **	Standard Care (*n* = 313) **	Unadjusted Estimate (95% CI)	Unadjusted *p* Value	Adjusted Estimate (95% CI)	Adjusted *p* Value
Chorioamnionitis ^a^	3 (0.95)	5 (1.60)	0.59 (0.14, 2.46)	0.470	0.56 (0.14, 2.28)	0.418
Induction of Labour ^a^	74 (23.42)	109 (34.96)	0.67 (0.52, 0.86)	0.002	0.66 (0.52, 0.85)	0.001
Antibiotics during Labour ^a^	147 (46.52)	137 (43.75)	1.06 (0.89, 1.26)	0.486	1.04 (0.88, 1.23)	0.629
Caesarean Section ^a^	73 (23.17)	74 (23.79)	0.97 (0.73, 1.29)	0.855	0.95 (0.72, 1.26)	0.713
Emergency Caesarean Section ^a^	41 (13.03)	45 (14.46)	0.90 (0.61, 1.33)	0.603	0.89 (0.60, 1.31)	0.560
Preterm Birth ^a^	23 (7.28)	20 (6.40)	1.14 (0.64, 2.03)	0.663	1.14 (0.64, 2.03)	0.669
Postpartum Haemorrhage ^a^	53 (16.84)	45 (14.43)	1.17 (0.81, 1.68)	0.408	1.16 (0.80, 1.67)	0.431
Perineal Trauma ^a^	184 (58.23)	189 (60.26)	0.97 (0.85, 1.10)	0.604	0.98 (0.86, 1.11)	0.728
3rd/4th Degree Perineal Trauma ^a^	9 (2.85)	5 (1.60)	1.78 (0.60, 5.25)	0.296	1.69 (0.57, 4.97)	0.344
Wound Infection ^a^	4 (1.27)	3 (0.99)	1.29 (0.29, 5.70)	0.740	1.45 (0.33, 6.39)	0.624
Postnatal Length Stay ^c^	1.87 (1.47)	1.88 (1.54)	0.99 (0.89, 1.11)	0.906	1.00 (0.89, 1.12)	0.951

^a^ Number and percentage of events, estimates are relative risks and 95% confidence interval. ^c^ Mean and standard deviation, estimates are relative risk ratios and 95% confidence intervals. ^f^ Adjusted analyses: all outcomes were adjusted for the stratification variable parity (0 vs. 1+), maternal age (continuous), maternal pre-pregnancy BMI (continuous) and SEIFA IRSD Quintile. ** = includes all women randomised who did not withdraw consent to use their data, and who did not suffer miscarriage or termination of pregnancy prior to 20 weeks gestation, or stillbirth.

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
