# Peer review of "A Randomised Trial to Optimise Gestational Weight Gain and Improve Maternal and Infant Health Outcomes through Antenatal Dietary, Lifestyle and Exercise Advice: The OPTIMISE Randomised Trial"

_nutrients, 2019, doi:10.3390/nu11122911_

Round 1

Reviewer 1 Report

The authors have examined the effect of dietary and exercise advice among pregnant women of normal BMI, on pregnancy and birth outcomes.

The study is done well and the data are clearly presented. In general, the study confirms several observations. Below I have made some specific points to address for the improvement of the manuscript.

Materials and Methods:

Line 71: Author mentioned that they have excluded subjects with diabetes (type or type 2) diagnosed prior to pregnancy. What if they were diagnosed during pregnancy (gestational diabetes)? Even though the women with gestational diabetes receive proper treatment, the condition can affect the weight of the baby.

Line 76: Better to mention the average gestational weeks at the time of 1st clinic visit.

Line 104: Explain SMART goal approach.

Line 108: Normally information relating to “weight gain during pregnancy” is one of the major component of standard antenatal care.

Line 111: Was the Gestational age at trial entry same/nearly same for everyone?

Line 126: the sentence looks incomplete as the next one is starting form a new number.

Line 139: Better to rewrite “length of stay” as “length of hospital stay”.; Why the “infant exclusively breast fed at hospital” mentioned under adverse outcomes for the infant?

Line 145: There may be some caesarean sections purely due to maternal requests without any underlying pathology or medical reason.  Those should be excluded.

Line 148: Author mentioned that each woman’s weight was recorded at trial entry and at 36 weeks’ gestation or nearest to birth. Then how the author assessed average weekly gestational gain?

Line 167: Total number of recruited women should be corrected as 645 instead of 624.

Discussion

Line 340: What is the reason for selecting only the women with normal BMI? This interventions may give good results for the women with higher BMI (Overweight and obese).

Conclusion

Line 390: Findings may not similar for the other BMI categories. Therefore cannot generalize these results for whole pregnant population and it’s difficult to say that “providing such an intervention in pregnancy is not advocated”. Instead author may recommend further studies including pregnant with all BMI categories.

Author Response

Reviewer 1:

Materials and Methods:

Line 71: Author mentioned that they have excluded subjects with diabetes (type or type 2) diagnosed prior to pregnancy. What if they were diagnosed during pregnancy (gestational diabetes)? Even though the women with gestational diabetes receive proper treatment, the condition can affect the weight of the baby.

We agree that gestational diabetes can influence fetal growth and infant birth weight. All women who were diagnosed with gestational diabetes after randomisation remained in the trial, consistent with an intention to treat analysis. As stated on line 121 of the manuscript, all women who were diagnosed with gestational diabetes received the appropriate clinical care (dietary modification, blood glucose monitoring and medication (insulin or metformin) as required).

Line 76: Better to mention the average gestational weeks at the time of 1st clinic visit.

Women were approached at the time of the first antenatal visit which could be between 10 and 20 weeks gestation, consistent with our inclusion criteria. Across South Australia more than 95% of women have their first antenatal appointment prior to 15 weeks gestation.

Line 104: Explain SMART goal approach.

The SMART goal approach includes setting goals that are Specific, Measurable, Achievable, Realistic, and Timely. Therefore, a SMART goal incorporates all of these criteria to increase the chances of goal achievement. A statement to this effect has been added to the manuscript (Line 105).

Line 108: Normally information relating to “weight gain during pregnancy” is one of the major components of standard antenatal care.

We agree that in many antenatal care settings, information about weight gain during pregnancy is part of standard care. However, as stated in the methods and referenced in the South Australian Perinatal Practice Guidelines, at the time of this randomised trial, standard antenatal care did not include provision of information to women about weight gain in pregnancy.

Line 111: Was the Gestational age at trial entry same/nearly same for everyone?

As presented in Table 1, the median gestational age at trial entry was 16.29 weeks, and similar between the two groups.

Line 126: the sentence looks incomplete as the next one is starting form a new number.

We have checked the manuscript and the sentence is grammatically correct, and consistent with journal formatting requirements.

Line 139: Better to rewrite “length of stay” as “length of hospital stay”; Why the “infant exclusively breast fed at hospital” mentioned under adverse outcomes for the infant?

Thank you for this suggestion – we have reworded this as length of hospital stay (Line 142).

We agree that this is not an adverse outcome, and have reworded to state infant not exclusively breast fed at hospital discharge (Line 142).

Line 145: There may be some caesarean sections purely due to maternal requests without any underlying pathology or medical reason.  Those should be excluded.

We agree that on occasion, maternal request for caesarean section occurs. However, this is a post-randomisation outcome, and consistent with an intention to treat analysis, these women remain in the analysis. 

Line 148: Author mentioned that each woman’s weight was recorded at trial entry and at 36 weeks’ gestation or nearest to birth…how was average weekly gestational gain assessed?

Average gestational weight gain was assessed by subtracting the weight at 36 weeks gestation or nearest to birth from the trial entry weight and divided by the number of completed weeks between the two measurements. 

Line 167: Total number of recruited women should be corrected as 645 instead of 624.

This statement relates to the sample size estimate. We have amended the statement to reflect the number of women required for recruitment, being 624 (Line 170). 

Discussion

Line 340: What is the reason for selecting only the women with normal BMI? This intervention may give good results for the women with higher BMI (Overweight and obese).

Our group has conducted multiple randomised trials of dietary and lifestyle interventions in pregnant women, including those who are overweight or obese. Please see our publications in the BMJ (Dodd et al 2014) and Lancet Diabetes & Endocrinology (Dodd et al 2019). Both of these randomised trials indicate that a similar dietary intervention was associated with improved maternal diet, but was not successful in improving clinical pregnancy outcomes, or reducing gestational weight gain.

Conclusion

Line 390: Findings may not be similar for the other BMI categories. Therefor, cannot generalize these results for whole pregnant population and it’s difficult to say that “providing such an intervention in pregnancy is not advocated”. Instead author may recommend further studies including pregnant with all BMI categories.

Please see our response to the above point – as indicated, we have conducted similar trials in women who are overweight or obese, and have demonstrated similar findings of improved dietary patterns, but no improvement in clinical pregnancy outcomes or reduced gestational weight gain. We consider our conclusion to be appropriate in light of the OPTIMISE trial findings, and the wider literature.

Reviewer 2 Report

Dodd et al described the results of the OPTIMISE RCT of dietary and lifestyle modification in prevention of excessive gestational weight gain and adverse outcomes. Previously, the authors published on the study design of the trial in detail. The current manuscript describes their findings. Over 600 normal weight pregnant women were randomized. The Lifestyle Advice group largely did not differ in outcomes compared to the Standard Care group, even though some efficacy evidence was shown in Table 3 of improvements in diet quality (although not physical activity). Some comments below:

Major comments

Abstract – please provide mean (SD) GWG OR the % excessive GWG by IoM of the 2 groups as it is focus of the introduction. Can remove this statement “Baseline characteristics of women in the two treatment groups were similar” to save space However, the main goal of the trial is macrosomia (bw >4kg)? In which case, the introduction should devote some text to describing the prevalence of macrosomia among NORMAL weight women and its importance as an outcome for long term maternal and neonatal morbidity? Unclear why the trial was atypically powered at 70%. That is low. However, the mean differences in proportions were <1% which means it would have been futile. There was >10% difference in inductions but not much difference in caesarean suggesting spontaneous labor was higher among the Lifestyle Advice group compared to standard care? That, taken together with the slightly earlier gestational age (but still delivered term at 39 weeks) is interesting. The authors mentioned the longer GA for standard care might have been from induction post-dates. Do the authors think this just spurious or is there any reason to think increased attempts at exercise (albeit not measurably different) or healthier diet played any role?

Minor comments

Abstract – incomplete/grammatically incorrect sentence? “The dietitian-led dietary and lifestyle intervention over the course of pregnancy, and was based on the Australian Guide to Healthy Eating.” Authors didn’t cite to their own methods paper published last year? Dodd et al. BMJ Open. 2018 Feb 20;8(2):e019583. doi: 10.1136/bmjopen-2017-019583 Unnecessary to mention fetal growth by ultrasound to just say it’s going into another manuscript. (lines 161-2 and again in lines 335-338).

Author Response

Reviewer 2:

Abstract – please provide mean (SD) GWG OR the % excessive GWG by IoM of the 2 groups as it is focus of the introduction.

Consistent with CONSORT reporting requirements for Randomised Trials, we have reported the primary outcome for the OPTIMISE trial in the abstract. The primary outcome was infant birth weight above 4kg. It is not considered best practice to selectively report secondary outcomes in the abstract. The secondary outcomes, including gestational weight gain have been extensively reported in the body of the manuscript.

Can remove this statement “Baseline characteristics of women in the two treatment groups were similar” to save space

Again, consistent with CONSORT reporting requirements for Randomised Trials, a statement has been included to reflect the comparability of both treatment groups at the time of randomisation.

However, the main goal of the trial is macrosomia (bw >4kg)? In which case, the introduction should devote some text to describing the prevalence of macrosomia among NORMAL weight women and its importance as an outcome for long term maternal and neonatal morbidity?

The introduction highlights firstly the risks of high gestational weight gain in women with normal BMI. The intervention was targeting gestational weight gain as a means to potentially reduce infant birth weight and improve pregnancy outcomes. The introduction then goes on to highlight the associations between excessive gestational weight gain, high infant birth weight, clinical pregnancy complications, and in the longer term poor maternal cardiovascular health and diabetes, as well as the association between both excessive gestational weight gain, high infant birth weight and child obesity. All of the included text refers to risks in women with normal BMI and gestational weight gain.

Unclear why the trial was atypically powered at 70%. That is low. However, the mean differences in proportions were <1% which means it would have been futile.

The power of 70% was a tradeoff between acceptable statistical power to detect clinically meaningful but plausible effects, and practical considerations of achievable sample size. Power in relation to continuous secondary outcomes was very high, so that any effect on infant weight is likely to have been detected.

Regarding the point about futility: we cannot know that an effect is too small to be meaningful before actually performing the study, so we respectfully disagree that the trial was futile. Note also that the difference in proportions of <1% is the observed difference; the 95% confidence interval around the point estimate gives a range of values which are also compatible with the data, hence we cannot conclude that this is the true population difference in proportions.

There was >10% difference in inductions but not much difference in caesarean suggesting spontaneous labor was higher among the Lifestyle Advice group compared to standard care? That, taken together with the slightly earlier gestational age (but still delivered term at 39 weeks) is interesting. The authors mentioned the longer GA for standard care might have been from induction post-dates. Do the authors think this just spurious or is there any reason to think increased attempts at exercise (albeit not measurably different) or healthier diet played any role?

We agree that this is an interesting finding. However, given the large number of statistical comparisons performed and both CS and induction of labour being secondary outcomes, it is highly likely that the findings are spurious and therefore should be interpreted with caution.

Abstract – incomplete/grammatically incorrect sentence? “The dietitian-led dietary and lifestyle intervention over the course of pregnancy, and was based on the Australian Guide to Healthy Eating.”

Thank you. This has been amended to state “The dietitian-led dietary and lifestyle intervention over the course of pregnancy was based on the Australian Guide to Healthy Eating.

Authors didn’t cite to their own methods paper published last year? Dodd et al. BMJ Open. 2018 Feb 20;8(2):e019583. doi: 10.1136/bmjopen-2017-019583

Thank you – this is an oversight and reference to our published protocol has been added.

Unnecessary to mention fetal growth by ultrasound to just say it’s going into another manuscript. (lines 161-2 and again in lines 335-338).

Thank you – the references to fetal growth by ultrasound have been removed as suggested.

Reviewer 3 Report

Manuscript “A randomised trial to optimise gestational weight gain and improve maternal and infant health outcomes through antenatal dietary, lifestyle and exercise advice: the OPTIMISE randomised trial” (Nutrients 649501).

The authors have extensive experience in research on nutrition in pregnancy. On this occasion, the objective of the randomized trial (OPTIMIZE) was to evaluate the effect of dietary and exercise advice among pregnant women with normal BMI, on pregnancy and childbirth outcomes. The primary outcome of the trial was the proportion of infants with birth weight > 4 kg. Their findings indicate that providing lifestyle advice during pregnancy to women with BMI within the normal range was associated with improvements in the maternal diet quality over the curse of pregnancy. However, despite improvements in the maternal diet, lifestyle advice was not associated with any difference in total gestational weight gain or the risk of weight gain below or above the IOM recommendations. There were no significant differences in clinical outcomes for women or their babies, including the risk of birth weight > 4 kg.

Comments and Suggestions for Authors:

The manuscript is an interesting study, but requires some considerations.

The biggest limitation of the study is that the participants in trial were predominantly white Caucasian, with less than half of women from areas of high social disadvantage. Furthermore, 75% of eligible women declined participation. These factors may limit the external validity and generalisability of the findings to other patient populations, as the authors assume.

It should be commented because infant SFTM and other maternal antenatal measures had between 20% - 40% missing data (line 187), since tuis rate is very high.

Probably, considering the BMI as a categorical variable, suppose a bias against its treatment as a continuous variable. In this last way, the BMI effect would have a continued effect from normality, overweight and obesity.

The authors admit that “Pre-specified secondary analyses identified some evidence of effect modification by maternal pre-pregnancy BMI, suggesting that the intervention may have been more effective in women with higher maternal BMI in reducing infant birth weight and head, abdominal and chest circumferences” (Line 328).  To achieve significant improvements in maternal and child health, generalized efforts must be made in the population to improve the health in general, and the weight specifically, of young people preconception. In this sense, the consideration that the study could be more important if carried out in a pre-gestational manner should be taken into account in the Discussion section.

It is noteworthy that among the reasons for exclusion, patients with pathological pregnancies who need to rest or who need to modify their diet or level of physical activity, other than pregestational diabetes type 1 or 2, are not considered (line 70). And likewise, that “women diagnosed with gestational diabetes remained in the study and were offered treatment with further dietary modification and metformin or insulin added as required to maintain appropriate glycaemic control” (line 121), without being excluded from the study?

Please find below a list of more specific issues that need to be addressed throughout the manuscript:

- Line 55 - Bibliographic citations [30], [31] should be linked as [30,31].

- Line 187 - The acronym SFTM, which appears for the first time in the text, should be described in parentheses (skinfold thickness measurement).

References should be reviewed. For example:

- Line 465 - Reference 18. The correct reference would be: Clin Nutr 2015, 34, 291-295.

- Line 507 - Reference 34. The correct reference would be: Acta ObstetGynecol Scand 2016, 95, 259-269.

- Line 585 - Reference 62. Add page: j3119.

Author Response

Reviewer 3:

The biggest limitation of the study is that the participants in trial were predominantly white Caucasian, with less than half of women from areas of high social disadvantage. Furthermore, 75% of eligible women declined participation. These factors may limit the external validity and generalisability of the findings to other patient populations, as the authors assume.

Thank you – we agree, as stated in our manuscript that these are indeed limitations to the generalisabilty and external validity of our findings.

It should be commented because infant SFTM and other maternal antenatal measures had between 20% - 40% missing data (line 187), since this rate is very high.

We agree that there is missing data for a range of secondary outcome variables. However, consistent with standard statistical practice, our methods included imputation to reduce the risk of bias that this may have introduced.

Probably, considering the BMI as a categorical variable, suppose a bias against its treatment as a continuous variable. In this last way, the BMI effect would have a continued effect from normality, overweight and obesity.

Apologies to this reviewer – it is not clear what is being asked.

The authors admit that “Pre-specified secondary analyses identified some evidence of effect modification by maternal pre-pregnancy BMI, suggesting that the intervention may have been more effective in women with higher maternal BMI in reducing infant birth weight and head, abdominal and chest circumferences” (Line 328).  To achieve significant improvements in maternal and child health, generalized efforts must be made in the population to improve the health in general, and the weight specifically, of young people preconception. In this sense, the consideration that the study could be more important if carried out in a pre-gestational manner should be taken into account in the Discussion section.

We agree that a focus on preconception care is essential, and this is a focus of ongoing future research studies.

It is noteworthy that among the reasons for exclusion, patients with pathological pregnancies who need to rest or who need to modify their diet or level of physical activity, other than pregestational diabetes type 1 or 2, are not considered (line 70). And likewise, that “women diagnosed with gestational diabetes remained in the study and were offered treatment with further dietary modification and metformin or insulin added as required to maintain appropriate glycaemic control” (line 121), without being excluded from the study?

Please see our response to a similar query above from reviewer 1 with regards to women who developed gestational diabetes during pregnancy remaining in the trial analyses.

Please find below a list of more specific issues that need to be addressed throughout the manuscript:

Line 55 - Bibliographic citations [30], [31] should be linked as [30,31].

Line 187 - The acronym SFTM, which appears for the first time in the text, should be described in parentheses (skinfold thickness measurement).

References should be reviewed. For example:

Line 465 - Reference 18. The correct reference would be: Clin Nutr 2015, 34, 291-295.

Line 507 - Reference 34. The correct reference would be: Acta ObstetGynecol Scand 2016, 95, 259-269.

Line 585 - Reference 62. Add page: j3119.

Thank you – these points have been addressed throughout the manuscript as suggested.

Reviewer 4 Report

I want to commend the authors of this study on the writing of their manuscript. 

I think there is a track changes comma in the abstract of the manuscript that should be accepted or removed. 

I think the addition of a figure with the time points of manipulations for the Lifestyle Advice Group would improve the manuscript.

Did the authors do any ethnicity comparisons? I see a majority of their sample population was Caucasian, is there any evidence on whether ethnicity plays a role on weight gain during pregnancy? An addition to the discussion may be helpful. 

Author Response

Reviewer 4:

I think there is a track changes comma in the abstract of the manuscript that should be accepted or removed. 

Thank you – this has been amended.

Did the authors do any ethnicity comparisons? I see a majority of their sample population was Caucasian, is there any evidence on whether ethnicity plays a role on weight gain during pregnancy?

As highlighted by this reviewer, the majority of our sample population were of Caucasian ethnicity. Due to the very small number of other ethnic groups, it was not considered reliable to conduct subgroup analyses. Furthermore, a subgroup analysis based on ethnicity was not pre-specified in our protocol or statistical analysis plan.